# Histopathological Aspects of the Prognostic Factors for Salivary Gland Cancers

**DOI:** 10.3390/cancers15041236

**Published:** 2023-02-15

**Authors:** Haruto Nishida, Takahiro Kusaba, Kazuhiro Kawamura, Yuzo Oyama, Tsutomu Daa

**Affiliations:** Department of Diagnostic Pathology, Faculty of Medicine, Oita University, Oita 879-5593, Japan

**Keywords:** salivary gland cancers, prognostic factors, histopathology, genetics, molecules

## Abstract

**Simple Summary:**

We describe the currently known histopathological aspects of the prognostic factors for salivary gland cancers and discuss the genetics or molecules used as diagnostic tools that might serve as treatment targets in the future.

**Abstract:**

Salivary gland cancers (SGCs) are diagnosed using histopathological examination, which significantly contributes to their progression, including lymph node/distant metastasis or local recurrence. In the current World Health Organization (WHO) Classification of Head and Neck Tumors: Salivary Glands (5th edition), malignant and benign epithelial tumors are classified into 21 and 15 tumor types, respectively. All malignant tumors have the potential for lymph node/distant metastasis or local recurrence. In particular, mucoepidermoid carcinoma (MEC), adenoid cystic carcinoma (AdCC), salivary duct carcinoma, salivary carcinoma, not otherwise specified (NOS, formerly known as adenocarcinoma, NOS), myoepithelial carcinoma, epithelial–myoepithelial carcinoma, and carcinoma ex pleomorphic adenoma (PA) are relatively prevalent. High-grade transformation is an important aspect of tumor progression in SGCs. MEC, AdCC, salivary carcinoma, and NOS have a distinct grading system; however, a universal histological grading system for SGCs has not yet been recommended. Conversely, PA is considered benign; nonetheless, it should be cautiously treated to avoid the development of metastasizing/recurrent PA. The aim of this review is to describe the current histopathological aspects of the prognostic factors for SGCs and discuss the genes or molecules used as diagnostic tools that might have treatment target potential in the future.

## 1. Introduction

In the current World Health Organization (WHO) Classification of Head and Neck Tumors: Salivary Glands (5th edition), malignant and benign epithelial tumors are classified into 21 and 15 tumor types, respectively (Table 1) [1]. 

The salivary gland comprises three major salivary glands (parotid, submandibular, and sublingual glands) and several minor salivary glands. Salivary gland cancers (SGCs) can arise from any salivary gland and have a morbidity rate of approximately 10–20% [2,3] (Bishop, J.A. et al. pp. 31–51). SGCs are diagnosed using histopathological examination, and histological findings can be considered the most valuable and distinct prognostic factors [2,3] (Bishop, J.A. et al. pp. 31–51). SGCs show various tumor types because a healthy salivary gland contains inner luminal/epithelial or acinar/mucous cells and outer basal/myoepithelial cells in the duct or the secretory part. Histopathological features correlate with tumor progression, including lymph node/distant metastasis or local recurrence. In 1986, Spiro reported that the significant prognostic factors were the site of origin, histologic subtype, histologic grading, and clinical stage [4]. However, some carcinomas show mild cytological atypia, making the evaluation of tumor invasion challenging. For all malignant tumors, including mucoepidermoid carcinoma (MEC), adenoid cystic carcinoma (AdCC), salivary carcinoma, and not otherwise specified (NOS) (formerly known as adenocarcinoma, NOS), validated grading systems exist; however, a universal histological grading system for SGCs has not been recommended [1]. Furthermore, novel genes or genetic components and proteins are being validated as diagnostic tools, and some of them may even serve as targets for new drugs. Herein, we summarize present SGC classifications with a focus on the prognostic factors and discuss new and potential prognostic factors from the histopathological viewpoint.

## 2. The Past and Present of SGCs

### 2.1. The General Histopathological Prognostic Factors for SGCs

#### 2.1.1. The Histological Types

The most prevalent malignancies are MEC and AdCC, and most SGCs have a distinct histological grade (Table 2). Histological diagnosis reflects biological behavior in several cases. The current WHO classification (5th edition) describes the grading system for MEC, AdCC, salivary carcinoma, and NOS [1]. Other tumors that show similar classification include carcinoma ex pleomorphic adenoma (CXPA) and intraductal carcinoma (IDC). The WHO classification detaches grades from tumor names since tumors with the same features of a cancer type do not necessarily have the same severity or aggressiveness, allowing for flexibility in describing tumors [1,2,5] (Bishop, J.A. et al. pp. 31–51). Each tumor’s histological features are subsequently discussed. 

#### 2.1.2. High-Grade Transformation (Dedifferentiation)

Dedifferentiation is a regression from a more differentiated to a less differentiated state (stem-cell-like). In particular, in a malignant tumor, a differentiated cell loses its specific form or function [6,7]. Histopathologically, dedifferentiation is observed as the abrupt transformation of a well-differentiated tumor into high-grade morphology (poorly differentiated or anaplastic/undifferentiated), lacking the original morphology (Figure 1) [8].

In the high-grade region, the tumor cells show anaplastic cells with large vesicular pleomorphic nuclei, prominent nucleoli, an increased mitoses/Ki67 labeling index, and necrosis [8,9]. The term “dedifferentiation” is occasionally used in malignant soft tissue tumors, such as dedifferentiated liposarcoma or dedifferentiated chondrosarcoma [10,11]. However, in SGCs, a malignant tumor is rarely replaced by a completely different histological morphology, and the original morphological features usually remain. Therefore, the term high-grade transformation is used to describe this phenomenon [8,12]. This transformation is reported not only in variable low-grade malignant tumors (acinic cell carcinoma, MEC, secretory carcinoma, hyalinizing clear cell carcinoma, myoepithelial carcinoma, epithelial–myoepithelial carcinoma, and polymorphous adenocarcinoma) but also in high-grade tumors, including AdCC [12,13,14,15,16,17,18,19,20,21]. Tumors with this finding have an even worse prognosis.

#### 2.1.3. Micropapillary Pattern

Invasive micropapillary carcinoma (IMPC) was first reported in the breast [22]. Neoplastic cell nests are uniformly distributed throughout a reticulated interstitium and exhibit a reverse polarity or “inside-out” growth pattern. This histological finding is observed in other malignant tumors of the urinary bladder, lung, stomach, colon, and bile duct [23,24,25,26,27]. IMPC, a tumor with a micropapillary pattern, frequently shows lymphatic invasion and lymph node metastasis, and its prognosis is very poor. Among SGCs, micropapillary salivary duct carcinoma (SDC) is the most prevalent; nonetheless, micropapillary AdCC and intraductal papillary mucinous neoplasm (IPMN) have also been reported (Figure 2) [21,28,29].

#### 2.1.4. Other Histologic Findings

Strong prognostic factors prevalent in many tumors include increased cellular atypia, perineural invasion, lymphovascular invasion, an increased mitoses/Ki67 labeling index, necrosis, local recurrence/distant metastasis, and poor surgical margin (Figure 3) [2,3,4,5] (Bishop, J.A. et al. pp. 31–51). 

For lymph node metastasis, important prognostic factors include the number of nodes, foci size, unilateral/bilateral involvement, extranodal extension, and stromal reaction [30,31,32]. Lombardi reported that intraparotid node metastasis implies an increased risk of lateral neck involvement and impact on the survival of patients with SGCs [33]. More specifically, the overall number (0 vs. 1–3 vs. ≥4) and diameter (<20 mm vs. ≥20 mm) of the node metastasis represent major prognostic factors for overall survival [33]. Additionally, for parotid gland carcinoma, facial nerve paralysis and tumor adhesion/immobility could be the predictive factors for high-grade SGCs [34].

### 2.2. Other Related Factors

Although not directly involved histopathologically, the most predictive factors for tumor recurrence are advanced age, male sex, large tumor size, and high clinical stage [2,35,36,37] (Bishop, J.A. et al. pp. 31–51). Concerning the site of tumor origin, small salivary glands have a high frequency of SGC occurrence, and the sublingual salivary gland has the highest frequency of malignancy [2,38] (Bishop, J.A. et al. pp. 31–51). 

## 3. Validated Grading Systems for Individual SGCs and Similar Diseases

This section discusses MEC, AdCC, salivary carcinoma, and NOS, which have validated grading systems in the WHO classification (5th edition), focusing on structural atypia, such as the presence or absence of a tumor nest/solid part and cellular atypia. Furthermore, we describe IDC and CXPA, which are classified by the presence or absence of tumor invasion and atypical cellular morphology. Moreover, metastasizing PA (MPA) is also discussed. 

### 3.1. Grading System in the WHO Classification

#### 3.1.1. Mucoepidermoid Carcinoma (MEC)

MEC is characterized by mucous, as we intermediate and epidermoid/squamoid tumor cells. However, in some tumors, intermediate cells are predominant. MEC forms cystic and solid growth patterns that are usually associated with *MAML2* rearrangement [2,39] (Bishop, J.A. et al. pp. 265–290); therefore, some reports describe the histological features for grading (Table 3, Figure 4) [2,39,40,41,42,43], which include structural atypia (cystic/solid component, border invasion pattern, lymphovascular/perineural invasion, and necrosis) and cytological atypia (nuclear anaplasia/pleomorphism and mitoses) [2,36,37,38,39,40,41,42,43]. According to these gradings, MEC should be graded as low, intermediate, or high. The higher the grade, the greater the possibility of metastasis or recurrence [2,39] (Bishop, J.A. et al. pp. 265–290).

The Armed Forces Institute of Pathology (AFIP) grading system was previously used; however, this system may not necessarily indicate the actual degree of some aggressive cases [2,40] (Bishop, J.A. et al. pp. 265–290); hence, the Brandwein system was introduced to classify these cases from the viewpoint of anaplasia [42]. Nevertheless, high-grade MEC is very rare, and there is no difference in outcome between low and intermediate grades using any grading system. Another study reported that mitosis and necrosis may be helpful in the classification of tumor grade [43].

#### 3.1.2. Adenoid Cystic Carcinoma (AdCC)

AdCC consists of two main cell types: ductal cells located in the inner part and myoepithelial cells located in the outer part of the duct. The ductal cells have eosinophilic cytoplasm and uniformly round nuclei, while the myoepithelial cells have a clear cytoplasm and hyperchromatic angular nuclei [2,44] (Bishop, J.A. et al. pp. 337–356). Perineural invasion is an AdCC hallmark, and genetically, it is characterized by *MYB* or related gene translocations. Typically, AdCC comprises pseudocysts and true glandular lumina. AdCC shows three growth patterns: tubular, cribriform, and solid [2,44] (Bishop, J.A. et al. pp. 337–356). Consequently, the following histological grading is used for its classification: tubular predominant as grade Ⅰ, cribriform predominant as grade Ⅱ, and solid predominant as grade Ⅲ (Figure 5) [2,44,45]. 

Although AdCCs with >30% solid components have been shown to be more aggressive, any solid tumor component may be a high-grade tumor, as described in the minAmax system [46,47,48]. Necrosis, marked pleomorphism, or high levels of mitoses are only seen in the solid pattern and are not utilized in the grading system [2] (pp. 337–356). 

#### 3.1.3. Salivary Carcinoma, NOS (Adenocarcinoma, Not Otherwise Specified (NOS), Formerly)

In the current WHO classification (5th edition), adenocarcinoma, NOS, is renamed as salivary carcinoma, NOS, and we have used the same term herein [49]. It includes the subtypes of oncocytic and intestinal-type adenocarcinoma. The term salivary carcinoma, NOS, should be used for tumors arising in major/minor salivary glands; however, this category is a heterogeneous spectrum of carcinomas showing ductal and/or glandular differentiation. It represents an exclusive diagnosis of otherwise defined salivary gland carcinoma entities (adenocarcinoma with nonspecific appearance) [2,49,50]. Due to differences in this carcinoma’s interpretation, the percentage or case numbers have varied in previous reports, and the pure entity adenocarcinoma, NOS, now accounts for approximately 10% of SGCs [36,51,52,53,54,55]. Adenocarcinoma, NOS, was considered a heterogeneous group of tumors; however, the strictly selected cases were considered as the pure group [56]. The histological grading system was based on a previous report published in 1982 by Spiro et al., and the tumors were classified as low-, intermediate-, or high-grade [2,49,54,55] (Bishop, J.A. et al. pp. 290–303). They defined anaplastic or high-grade lesions as grade Ⅲ, which are arranged in close clumps and broad bands composed of small glandular tumor cells. The tumor cell nests are separated by collagen connective tissue stroma, similar to that in seen during scirrhous formation. In addition, they are divided into low- (grade Ⅰ) or intermediate-grade (grade II); the low-grade variant shows no stromal invasion, whereas the intermediate grade shows definite stromal infiltration. Grade Ⅱ lesions have prominent sheets or cords of some polymorphic glandular cells with a pale cytoplasm [54]. Although this classification reflected the tumor prognosis at that time, it is still somewhat reasonably useful today; however, it should be revised owing to the differences in current diagnostic criteria.

### 3.2. Carcinoma ex Pleomorphic Adenoma (CXPA)

CXPA is an epithelial and/or myoepithelial malignancy arising in a primary or recurrent pleomorphic adenoma (PA) [2,57] (Bishop, J.A. et al. pp. 374–400). Reflecting the presence of PA, the typical clinical presentation is a long-term painless mass with recent rapid progression or previous PA diagnosis. PA presents as ductal and myoepithelial cells arranged in bilayered tubular structures, while the stroma is typically mucoid, myxoid, hyalinized, or chondroid. Usually, the transition from PA to the malignant component is distinct; however, some cases may have heavy stromal/hyalinized collagen bundles or chondroid/myxoid stroma only. The common malignancies are SDC, epithelial–myoepithelial carcinoma, salivary carcinoma, NOS, and myoepithelial carcinoma; in contrast, carcinosarcoma is rare [58]. Most carcinosarcomas arise from PA through the intraductal or myoepithelial pathway, the multistep adenoma–carcinoma–sarcoma sequence [59]. The relevant observations are the histological subtype/grade, the proportion of carcinoma (>50%), and the extent of invasion [2,58,59,60] (Bishop, J.A. et al. pp. 374–400). The malignant component progresses from an encapsulated neoplasm to extracapsular invasion. The term “encapsulated” has been rephrased by various alternatives, such as “intracapsular”, “in situ”, “preinvasive”, “intramural”, or “noninvasive”; however, the term intracapsular is preferred. CXPA is subclassified based on the extent of invasion beyond the PA as follows: intracapsular, minimally invasive (the carcinoma invades <4–6 mm beyond the PA borders), and invasive (invasion beyond the PA capsule ≥6 mm) [57]. However, evaluating the fibrous capsule is occasionally challenging because the tumor forms the capsule or the capsule outline is vague, especially in primary minor salivary glands. Several specimen preparations are needed for a precise diagnosis. 

### 3.3. Intraductal Carcinoma (IDC)

IDC is a salivary gland malignancy located entirely or predominantly intraductally [2,61] (Bishop, J.A. et al. pp. 461–475). It has papillary, cribriform, and solid structures mimicking atypical ductal hyperplasia or ductal carcinoma in situ of the breast [2,61] (Bishop, J.A. et al. pp. 461–475). It shows four subtypes based on the tumor cells: intercalated duct, apocrine, oncocytic, and mixed IDC; hence, tumors in this category are heterogeneous. Usually, intercalated duct and oncocytic IDC are low-grade, whereas apocrine and mixed IDC are low- or high-grade [2,61] (Bishop, J.A. et al. pp. 461–475). Independent of tumor cell subtype, pure IDC behaves indolently, although invasive carcinomas ex-IDC (arising from IDC) can behave aggressively [62,63]. Therefore, these tumors may be reassessed as “IDC, noninvasive (low-grade IDC)”, “IDC, noninvasive (high-grade IDC)”, or “IDC, invasive”, similar to intraductal papillary mucinous neoplasm (IPMN) of the pancreas or CXPA of the salivary gland.

### 3.4. Metastasizing Pleomorphic Adenoma (MPA)

PA is a benign tumor composed of benign ductal and myoepithelial cells with a chondromyxoid or sclerosing fibrous component in the background [2,64] (Bishop, J.A. et al. pp. 536–543). In peculiar cases, benign-looking PA metastasizes to the bone, lung, and neck lymph nodes with some local recurrence [65,66]. Knight et al. reported that 41 patients (80.4%) with MPA were alive at the time of 1-year follow-up. However, survival was poor, and 17.6% (9/51 cases) died from MPA [66]. Although the term includes “PA”, considering the clinical course, it was managed at least as a low-grade malignancy. There are no histological or molecular features to predict metastasis, and it is not distinguishable from a benign tumor at the primary site [66,67]. Thus, first-time surgery for PA must be performed cautiously if the patient is young, the PA is multinodular, or there is a tumor rupture. Furthermore, a postoperative status of incomplete tumor excision, incomplete pseudocapsule, extracapsular extension (microscopic pseudopods or skip/satellite lesion beyond the pseudocapsule), or poor margin should also be considered due to the possibility of recurrent PA [68,69,70,71]. 

## 4. Future Perspectives on SGCs

This section discusses SGC-specific genes or characteristic proteins, including their immunohistochemistry. These have emerged as diagnostic tools in recent years and could be clinicopathological predictors of SGCs. Although many drugs are not included in the usual regimens, drug-targetable proteins and genes, such as hormone receptors in breast cancers, have been shown to alter SGCs prognosis. Although various methods are used to investigate SGCs or target genes, in several cases, immunohistochemistry (IHC), reverse transcription polymerase chain reaction (RT-PCR), fluorescence in situ hybridization (FISH), and Sanger sequencing/next-generation sequencing (NGS) are commonly used. IHC, RT-PCR, and FISH detect protein expression, fusion genes and gene translocation, and gene translocation and amplification, respectively. Furthermore, Sanger sequencing detects point mutations or minor genetic alterations, and NGS can be used for whole-genome sequencing; nevertheless, IHC is the most commonly performed investigation, as it is economic and convenient. Recently, targeted therapies have been developed to target the signaling pathways involving the molecular signatures detected by these techniques. For example, if the detected oncogene signature is *VEGF/ANG2*, *VEGFR/FGFR/PDGFR*, *HER2*, *EGFR*, or *TrkB/BDNF*, the targeted therapy for MEC is sorafenib, nintedanib, trastuzumab, lapatinib, or ANA-12, respectively [72]. 

### 4.1. Genetics as a Diagnostic Tool

SGCs are heterogeneous tumors, and genetics aids in understanding the molecular biology of each tumor. All malignant tumors listed in the WHO classification (5th edition) and related genes are summarized in Table 4 [1,2,73,74,75,76,77,78] (Bishop, J.A. et al. pp. 31–51). 

These gene alterations are used as diagnostic tools and represent the tumor’s specific characteristics. Some of these genes are druggable, and HER2 or tropomyosin receptor kinase (TRK) inhibitors have been used to treat salivary duct or secretory carcinoma, respectively [76,77,78]. Some of these drugs significantly change patients’ prognoses, and they are discussed in the following section. 

### 4.2. Druggable Genes and Proteins (Including Drug Repositioning/Drug Repurposing)

#### 4.2.1. Human Epidermal Growth Factor Receptor 2 (HER2)

*HER2* is a proto-oncogene that is expressed or overexpressed in a variety of epithelial malignancies, including, breast, stomach, colon, rectum, biliary tract, and lung cancer [79,80,81,82,83,84,85,86,87]. Its overexpression is associated with *HER2* gene amplification or mutation. That is, the *HER2* gene is amplified in 20% to 25% of primary breast cancers. Accordingly, HER2 inhibitors are used to treat HER2-positive breast and gastric cancers [80]. In SGCs, the overall frequency of HER2 overexpression is 17% and is predominantly seen in SDCs [88]. Other HER2-high expression tumors are CXPA, adenocarcinoma, NOS, squamous cell carcinoma, and MEC [89]. HER2-positive tumors, for example, breast carcinoma, have been treated with trastuzumab (Herceptin), the use of which is expanding to gastric or colon cancers [81,82]. Recently, its use has been explored for the management of SDC, urothelial carcinoma, bile duct adenocarcinoma, etc. [83,84,85,86,87]. Trastuzumab treatment for HER2-positive patients is correlated with good response and long-term survival [84]. Notably, an advanced anti-HER2 antibody, trastuzumab deruxtecan (T-Dxd, Enhertu) was developed, which is used for HER2-low breast cancers (HER2 IHC score of 1+ or 2+ without gene amplification) [90]. T-Dxd is an antibody–drug conjugate combination of trastuzumab and topoisomerase I inhibitor, which implies that its use could be expanded in the future to treat a variety of tumors, including HER2-positive SDCs [91,92].

#### 4.2.2. Androgen Receptor (AR)/NK3 Homeobox 1 (NKX3.1)

AR expression is mainly characteristic of SDC among all SGCs [2,87] (Bishop, J.A. et al. pp. 475–497). Owing to AR copy number gain, ligand-independent splice variants, and mutations, AR is overexpressed in typical SDCs [2] (pp. 475–497). Some SDCs are also positive for NKX3.1, α-methylacyl-CoA racemase (AMACR), and prostatic acid phosphatase (PAP), and the positivity of these androgen hormone-related proteins was previously reported in prostatic cancer [93,94]. Hence, androgen deprivation therapy is part of the standard of care for advanced and metastatic prostate cancer [95]. Similar to that for prostatic cancer, androgen deprivation therapy has been performed, and some reports have stated that it is effective for SDC patients [96]. However, the prognosis of patients with AR-, AMACR-, or PAP-negative SDC remains poor [87,93]. In particular, AR negativity is associated with significantly worse overall survival, as splice variants and increased gene copy number may reduce the drug response and increase therapeutic resistance [97,98].

#### 4.2.3. Protein Receptor Kinase/Protein Kinase

Various genetic alterations have been reported for SGCs, including some oncogenic driver alterations (for example, *EGFR* mutation and *ALK* translocation; Table 4) [1,2,99] (Bishop, J.A. et al. pp. 31–51). These alterations accelerate tumorigenesis. For example, *RET* or *MET* acts on tumors with activating alterations as proto-oncogenes, such as point mutations or fusions. Therefore, these alterations are an easy therapeutic target. Alterations in *MET* and *RET* have been reported in 1.2% and 0.8% of non-small cell lung cancers, respectively [100,101,102]. Meanwhile, MET and RET inhibitors exhibit high efficacy rates and good tolerability [100,101,102,103]. Particularly for SGCs, the use of tyrosine kinase inhibitor (TKI)/TRK inhibitor is expected to be beneficial because these protein-coding gene alterations are detectable. Representative TKIs include those against vascular endothelial growth factor receptors (VEGFRs), fibroblast growth factor receptors (FGFRs), platelet-derived growth factor receptors (PDGFRs), etc. [104]. These TKIs are related to cell regulation and survival, as they influence angiogenesis and lymphangiogenesis. VEGFR inhibitors have been successfully used for the treatment of lung, stomach, liver, and kidney cancer [105]. Additionally, PDGFR inhibitors target gastrointestinal tumors, glioblastomas, sarcomas, leukemias, and dermatofibrosarcoma protuberans [106]. TRK is encoded by the *NTRK* gene family (*NTRK1, NTRK2*, and *NTRK3*) [107]. This proto-oncogene is responsible for cancer cell transformation, tumor cell proliferation, migration, and invasion. NTRK expression is detectable in approximately 90% of certain cancer types, including secretory breast carcinoma, secretory carcinoma in the salivary gland, and congenital infantile fibrosarcoma, while it is reported in less than 1% of common cancers such as non-small cell lung, colorectal, thyroid, and salivary gland cancers [108]. Currently, numerous protein inhibitors are being developed for various tumors and are expected to be effective against SGCs, as many carry genetic alterations associated with tumorigenesis [59,77,81,109,110,111]. Among SGCs, the overall response rates to protein inhibitors and disease control rates (0–46.2% and 59.7–100.0%, respectively) are similar to those reported in chemotherapy trials [112].

#### 4.2.4. Tumor-Infiltrating Lymphocytes (TILs)/Immunotherapy-Related Proteins

SGCs have shown lymphocytic infiltrations with or without lymphoid follicles, and certain cases are called tumor-associated lymphoid proliferation (TALP) [113]. This may be misdiagnosed as lymph node metastasis by pathologists if they are not aware; nonetheless, its relationship with the prognosis remains unclear [113]. In addition, the special SGC type, lymphoepithelial carcinoma, presents non-keratinizing, poorly differentiated squamous-cell-like carcinoma with a predominant lymphoid stroma [114]. Most cases show Epstein–Barr virus (EBV) infection, and this type of tumor has a relatively good prognosis [115]. Moreover, in a tumor-immune environment, TILs are related to the prognosis of breast cancer [116]. Although reports are conflicting, the number of TILs is associated with a better prognosis. Moreover, the therapeutic effect is frequently correlated with the number of TILs and tumor mutation burden (TMB) in many tumors, such as malignant melanoma, colon cancer, pancreatic cancer, or biliary tract cancer [117,118,119,120]. TMB is the total number of DNA alterations in cancer cells and is measurable using NGS. Moreover, mismatch repair deficiency (dMMR) or microsatellite instability (MSI-H) causes high TMB; colorectal cancer includes two subtypes (Lynch syndrome and sporadic MSI-H cancer) [121]. Approximately 20% of colon cancer patients are MSI-H, and approximately 3% of patients with MSI-H colon cancer are diagnosed with Lynch syndrome [122]. The US Food and Drug Administration approved immune checkpoint inhibitors for metastatic MSI-H colon cancer and solid tumors with dMMR/MSI-H [123]. In addition, anticytotoxic T-lymphocyte-associated antigen 4 (CTLA-4) agents and anti-PD-1 agent combination therapy have exhibited acceptable antitumor efficacy in MSI-H/dMMR metastatic colorectal cancers [123]. However, research on TILs related to global SGCs showed no significant difference in CD8^+^ TIL density between disease-free survival and overall survival [124]. This may be owing to the limitation of not treating the tumors individually or the low case numbers. Hence, further examination is required. 

Programmed death (PD)-1/programmed death-ligand 1 (PD-L1) exists in the tumor microenvironment. Immunotherapy can kill tumor cells by activating antitumor immunity against tumor antigens. In particular, PD-1 is the most important receptor responsible for activating T cells and mediating immunosuppression. However, PD-L1 is also involved in PD-1-related pathways, leading to the induction of T-cell apoptosis or anergy [125]. The PD-1/PD-L1 pathway is the most notable checkpoint inhibitor pathway. Moreover, CTLA-4 is also the immune checkpoint target in clinical practice for many tumors [125,126]. When CTLA-4 translocates to the cell surface, CTLA-4 mediates inhibitory signaling into the T cell and arrests cell proliferation and activation [127]. Hence, anti-CTLA-4 agents are expected to exhibit beneficial therapeutic effects. In particular, a monoclonal antibody against CTLA-4 effectively amplifies immune stimulation and boosts tumor annihilation [128]. This antibody has been applied for the treatment of non-small or small cell lung cancer, renal cell carcinoma, urothelial carcinoma, pancreatic cancer, gastric cancer, and malignant melanoma [128]. However, limited data are available regarding the therapeutic potential of immune checkpoint inhibitors for SGCs, especially MEC and AdCC [129]. In particular, PD-L2 may be an important biomarker in SGCs (for example, MEC, AdCC, and SDC) [126,130]. 

#### 4.2.5. Other Targetable Genes and Proteins

In addition, although rare, the following tumors have specific proteins or gene alterations involving tumorigenesis: (i) nuclear protein in testis (NUT); NUT carcinoma, (ii) subfamily of ATP-dependent chromatin remodeling complexes SWI/SNF (switch/sucrose non-fermentable); SWI/SNF-related, matrix-associated, actin-dependent regulator of chromatin, subfamily B, member 1 (SMARCB1)-deficient high-grade transformed/dedifferentiated acinic cell carcinoma, (iii) BRAF; IPMN, (iv) neuroendocrine granules; small cell neuroendocrine carcinoma “Merkel type” (SNECM), large cell neuroendocrine carcinoma, and (v) salivary gland carcinoma with viral infection (human papillomavirus, EBV, polyomaviruses, and so on) [131,132,133,134,135,136,137,138,139,140,141,142]. These unique proteins or gene alterations might be related to tumorigenesis and may therefore represent novel therapeutic targets. 

## 5. Conclusions

Herein, we discussed prognostic factors, focusing on the histopathological findings for SGCs, and described the current scenario and future perspectives. The interaction between clinicians and pathologists is essential since the pathological report includes many prognostic factors for patients and should be read carefully. Furthermore, genetics and molecular pathology are continuously advancing. Thus, novel information is continuously emerging, requiring further exploration.

## Figures and Tables

**Figure 1 cancers-15-01236-f001:**
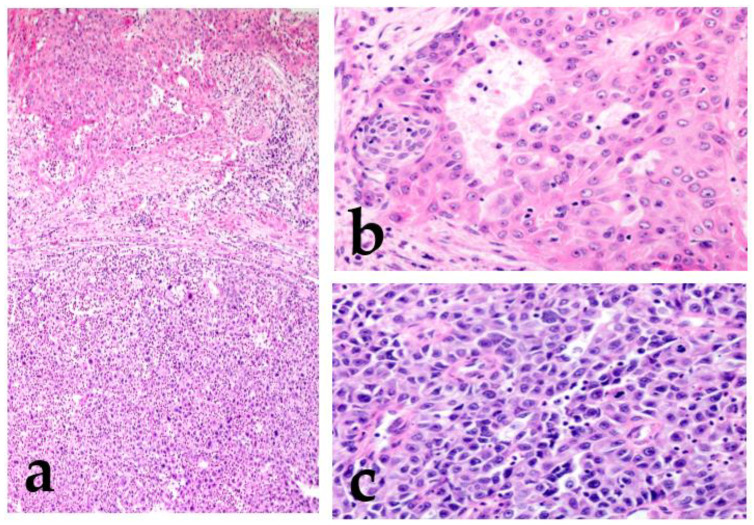
Mucoepidermoid carcinoma (MEC) with high-grade transformation. There is intermediate-grade MEC in the upper region, and the lower region shows high-grade transformation (**a**). Intermediate-grade MEC forms glandular or solid structures and consists mainly of intermediate cells with moderate atypia (increased nuclear size with an obvious nucleus) (**b**). The high-grade region shows undifferentiated features, and the cells lack the original morphology (**c**).

**Figure 2 cancers-15-01236-f002:**
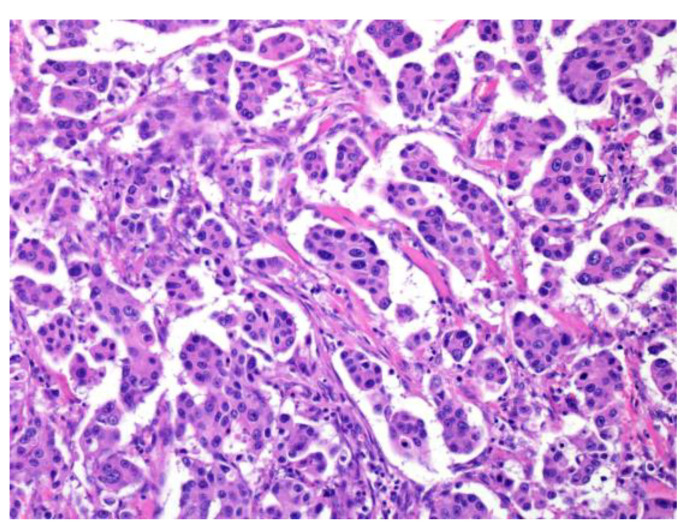
Micropapillary salivary duct carcinoma (SDC). The tumor forms many micropapillary nests that consist of eosinophilic cytoplasm and irregular nuclear-like SDC.

**Figure 3 cancers-15-01236-f003:**
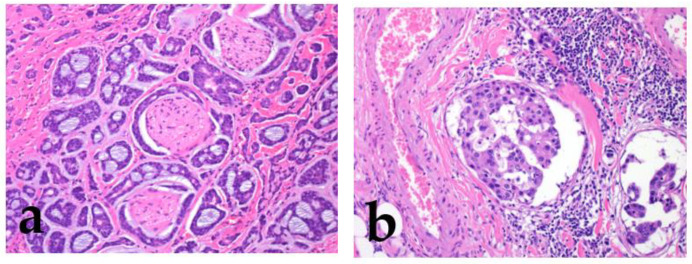
Perineural invasion and lymphatic invasion. Adenoid cystic carcinoma shows frequent perineural invasion (**a**), and salivary duct carcinoma shows lymphatic invasion (**b**).

**Figure 4 cancers-15-01236-f004:**
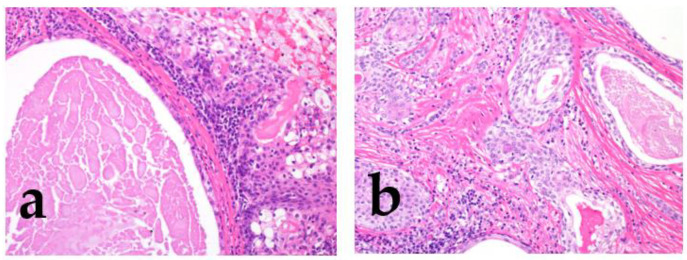
The morphological features of low/intermediate mucoepidermoid carcinoma (MEC). The tumor forms cystic (**a**; low grade) and solid growth patterns (**b**; intermediate). In both cases, tumor cell atypia is mild, but the intermediate category shows slightly different nuclear sizes with mitosis.

**Figure 5 cancers-15-01236-f005:**
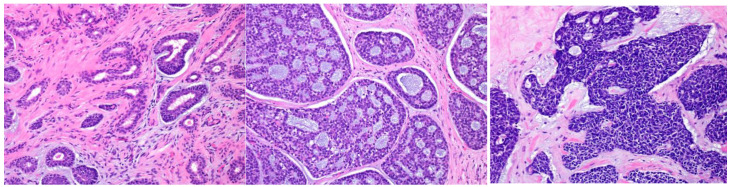
The histological grade of adenoid cystic carcinoma (AdCC). The tumor shows tubular (**left**), cribriform (**middle**), and solid (**right**) growth patterns.

**Table 1 cancers-15-01236-t001:** WHO Classification of Head and Neck Tumors: Salivary Glands (5th edition).

Benign Epithelial Tumours	Malignant Epithelial Tumours
Pleomorphic adenoma	Mucoepidermoid carcinoma
Basal cell adenoma	Adenoid cystic carcinoma
Warthin tumour	Acinic cell carcinoma
Oncocytoma	Secretory carcinoma
Salivary gland myoepithelioma	Microsecretory adenocarcinoma
Canalicular adenoma	Polymorphous adenocarcinoma
Cystadenoma of salivary gland	Hyalinizing clear cell carcinoma
Ductal papillomas	Basal cell adenocarcinoma
Sialadenoma papilliferum	Intraductal carcinoma
Lymphadenoma	Salivary duct carcinoma
Sebaceous adenoma	Myoepithelial carcinoma
Intercalated duct adenoma and hyperplasia	Epithelial-myoepithelial carcinoma
Striated duct adenoma	Mucinous adenocarcinoma
Sclerosing polycystic adenoma	Sclerosing microcystic adenocarcinoma
Keratocystoma	Carcinoma ex pleomorphic adenoma
	Carcinosarcoma of the salivary glands
**Mesenchymal tumours specific to the salivary glands**	Sebaceous adenocarcinoma
Sialolipoma	Lymphoepithelial carcinoma
	Squamous cell carcinoma
	Sialoblastoma
	Salivary carcinoma, NOS and emerging entities

**Table 2 cancers-15-01236-t002:** Native histopathological stratification of salivary gland malignancies.

Low-Grade Malignancy	Intermediate Malignancy	High-Grade Malignancy	Variable Grade
Acinic cell carcinoma	Myoepithelial carcinoma	Salivary duct carcinoma	Mucoepidermoid carcinoma
Basal cell adenocarcinoma	Sebaceous adenocarcinoma	Squamous cell carcinoma	Adenoid cystic carcinoma
Epithelial-myoepithelial carcinoma	Lymphoepithelial carcinoma	Small cell carcinoma	Salivary carcinoma, NOS
Secretory carcinoma		Large cell neuroendocrine carcinoma	Intraductal carcinoma
Polymorphous adenocarcinoma		Large cell undifferetiated carcinoma	Carcinoma ex pleomorphic adenoma
Hyalinizing clear cell carcinoma		Carcinosarcoma	
Mucinous adenocarcinoma		Salivary gland carcinomas with high-grade transformation	
Microsecretory adenocarcinoma			
Sclerosing microcystic adenocarcinoma			
Sialoblastoma			
(Metastasizing pleomorphic adenoma)			

**Table 3 cancers-15-01236-t003:** Comparison of the mucoepidermoid carcinoma grading systems.

Comparison of Mucoepidermoid Carcinoma Grading Systems			
Feature	AFIP [7,8]	Brandwein [9]	Katabi [10]
Cysts/Architecture	2 (<20% cystic)	2 (<25% cystic)	LG: predominantly cystic
			IG/HG: predominantly solid
Border/Invasive Front	n/a	2 (small nests & islands)	LG: circumscribed
			IG/HG: infiltrative
Necrosis	3	3	LG/IG: absent
			HG: present
Nuclear Anaplasia/Pleomorphism	4	2	LG/IG: not significant
Lymphovascular Invasion	n/a	3	n/a
Perineural Invasion	2	3	n/a
Mitoses	3 (4/10 HPF)	3 (5/10 HPF)	LG: 0–1/10 HPF
			IG: 2–3/10 HPF
			HG: 4+/10 HPF
Bony Invasion	n/a	3	n/a
Low Grade (LG)	0–4	0	Qualitative Assessment
Intermediate Grade (IG)	5–6	2–3	
High Grade (HG)	7–14	4–16	
LG: low grade; IG: intermediate grade; HG: high grade; n/a: not applicable; AFIP: Armed Forces Institute of Pathology			
If a pathologic feature is present, relevant points are assigned as listed in the table. Final grade is given by sum of points.			

**Table 4 cancers-15-01236-t004:** Comparison of histological diagnoses and gene alterations.

Tumour Type	Chromosomal Region	Gene Alterations
Mucoepidermoid carcinoma	t(11;19) (q21;p13)	CRTC1::MAML2
	t(11;15) (q21;q26)	CRTC3::MAML2
	9p21.3	CDKN2A deletion
Adenoid cystic carcinoma	6q22-23	MYB fusion/activation/amplification
	8q13	MYBL1 fusion/activation/amplification
	9q34.3	NOTCH mutations
Acinic cell carcinoma	9q31	NR4A3 fusion/activation
	9q31.1	MSANTD3 fusion/amplification
Secretory carcinoma	t(12;15) (p13;q25)	ETV6::NTRK3 fusion
	t(12;10) (p13;q11)	ETV6::RET fusion
	t(12;7) (p13;q31)	ETV6::MET fusion
	t(12;4) (p13;q31)	ETV6::MAML3 fusion
	t(10;10) (p13;q11)	VIM::RET fusion
Microsecretory adenocarcinoma	t(5q14.3) (18q11.2)	MEF2C::SS18 fusion
Polymorphous adenocarcinoma		
Classic subtype	14q12	PRKD1 mutations
Cribriform subtype	14q12	PRKD1 fusions
	19q13.2	PRKD2 fusions
	2p22.2	PRKD3 fusions
Hyalinizing clear cell carcinoma	t(12;22) (q21;q12)	EWSR1::ATF1 fusions
		EWSR1::CREB1 fusions
		EWSR1::CREM fusions
Basal cell adenocarcinoma	16q12.1	CYLD mutations
		CTNNB1 mutation
Intraductal carcinoma		
Intercalated duct subtype	10q11.21	RET fusions
		TRIM27::NCOA4 fusions
Apocrine subtype	3q26.32	PIK3CA mutations
	11p15.5	HRAS mutations
Salivary duct carcinoma	17q21.1	HER2 amplification
	8p11.23	FGFR1 amplification
	17p13.1	TP53 mutation
	3q26.32	PIK3CA mutation
	11p15.5	HRAS mutation
	Xq12	AR copy gain
	10q23.31	PTEN loss
	9p21.3	CDKN2A loss
Myoepithelial carcinoma	8q12	PLAG1 fusions
	t(12,22) (q21;q12)	EWSR1::ATF1 fusions
Epithelial-myoepithelial carcinoma	11p15.5	HRAS mutations
		PLAG1 fusion
		HMGA2 fusion
Mucinous adenocarcinoma	14q32.33	AKT1 p.E17K mutations
	17p13.1	TP53 mutations
Sclerosing microcystic adenocarcinoma	1p36.33	CDK11B mutation
Sebaceous adenocarcinoma	2p21	MSH2 loss
Carcinosarcoma	none specific	
Lymphoepithelial carcinoma	Not reported	
Squamous cell carcinoma	Not reported	
Sialoblastoma	Not reported	
Carcinoma ex pleomorphic adenoma	8q12	PLAG1 fusions/amplification
	12q13-15	HMGA2 fusions/amplification
	17p13.1	TP53 mutations
(Pleomorphic adenoma)	8q12	PLAG1 fusions/amplification
	12q13-15	HMGA2 fusions/amplification

## Data Availability

The data presented in this study are available upon request from the corresponding author.

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
