# Peer review of "Histopathological Aspects of the Prognostic Factors for Salivary Gland Cancers"

_cancers, 2023, doi:10.3390/cancers15041236_

Round 1
Reviewer 1 Report
This is a very well written review article focusing on the prognostic factors in salivary gland cancers (SGC). The authors properly discuss new and potential prognostic factors from the histopathological viewpoint. Validated grading systems for individual SGCs are available for mucoepidermoid carcinoma (MEC) adenoid cystic carcinoma (AdCC), and salivary carcinoma, NOS (adenocarcinoma, NOS), which have been included in the WHO classification (5th edition). Furthermore, the reviw focuses on high-grade transformation of SGCs, and it also describes intraductal carcinoma and carcinoma ex PA, which are classified by the presence or absence of tumor invasion.
I do not have any major critical comments, just a recommendation.
1. In the section discussing the prognostic factors and druggable genes and proteins, more attention should be given to NTRK and RET inhibitors.
2. NOTCH gene mutation has been shown important for prognosis of AdCC, and it should be discussed with availability of drugs targeting NOTCH signaling pathway.
3. Due to the development Myb-targeted inhibitors and an ongoing clinical trial of MYB-targeted cancer vaccine therapy, MYB is becoming an increasingly attractive therapeutic target for adenoid cystic carcinoma.
Author Response
Dear Reviewer 1:
Thank you for your review and positive feedback. Please find below our point-by-point response to each comment. All revisions in the manuscript have been highlighted in yellow.
Comment: In the section discussing the prognostic factors and druggable genes and proteins, more attention should be given to NTRK and RET inhibitors.
Response: Thank you for your recommendation. To address your comment, we have added RET and TRK inhibitors to the section.
Comment: NOTCH gene mutation has been shown important for prognosis of AdCC, and it should be discussed with availability of drugs targeting NOTCH signaling pathway.
Comment: Due to the development Myb-targeted inhibitors and an ongoing clinical trial of MYB-targeted cancer vaccine therapy, MYB is becoming an increasingly attractive therapeutic target for adenoid cystic carcinoma.
Response: Thank you for these comments and suggestions. We have referenced the MYB and NOTCH signaling pathways for AdCC in the “Other targetable genes and proteins” section.

Reviewer 2 Report
Based on the Title (“Histopathological aspects of the prognostic factors for salivary gland cancers”) and the Aim of the study (“…and discuss new and potential prognostic factors from the histopathological viewpoint”) this narrative review would be expected to link histopathological features of salivary gland carcinomas (SGCs) with tumor prognosis. However, it becomes clear from the Simple Summary (“and discuss the genetics or molecules used as diagnostic tools, which might have the potential of the treatment target for the future”) that the contribution of genetics in the diagnosis and treatment of SGCs will be also presented. In fact, “Future Perspectives on SGCs” that constitute a considerable part of the review is dedicated to the role that techniques other than histopathology may play in the diagnosis of SGCs (“Genetics as diagnostic tools”), or in the detection of possible drug targets in those lesions. Finally, the Aim states “We summarize the present SGCs classification…” This should not be included in the Aim as the presentation of the current classification of SGCs (in fact only of them, not all salivary gland tumors as in the manuscript) it is a prerequisite for the review.
For the above-mentioned reasons, the Title and Aim, or the content of the manuscript should be changed accordingly. In addition, a review should reflect the opinion of the authors, after the critical construction of available information. In most of the manuscript, it seems that they only present and summarize the information included in the 5th WHO classification.
The information included in the section “Future Perspectives on SGCs” is interesting but should be restructured in case that it should be included in a revised version of the manuscript. I.e., how those genetic alterations have been utilized in the diagnosis/treatment of other malignancies, how common are in SGCs and is specific subtypes, information on what has been done in the treatment of SGCs. This structure should be followed in all paragraphs.
In addition, due to numerous grammatical or syntaxis errors, the manuscript lacks consistency and is cannot be easily understood.
Some examples:
· “The salivary gland comprises three major salivary glands (parotid, submandibular, and sublingual glands) and minor salivary glands”. Please rephrase.
· “Salivary gland cancers (SGCs) arise from any salivary glands…”. SGCs (why not adenocarcinomas?) arise from any salivary gland (not plural form). Please rephrase.
· “…the morbidity is approximately 10 to 20%”. What those percentages stand for, incidence, prevalence? Please specify.
· “SGCs show various tumor types”. What do you mean? Please rephrase.
· “…due to the normal salivary gland consisting of inner luminal/epithelial or acinar/mucous cells and outer basal/myoepithelial cells in the duct or the secretory part.” Although I can think of what the authors intent to state, the sentence should be restructured.
· “The histopathological features correlate with the tumor progression, including lymph node/distant metastasis or local recurrence.” What are those “histopathological features”? Do you mean the histopathological subtypes?
· “A relatively long time ago, Spiro reported that the significant prognostic factors were the site of origin, histologic subtype, histologic grading, and clinical stage [4]. However, some carcinomas show mild cytological atypia, or evaluating the tumor invasion is challenging.” It is profound that cytological atypia and tumor invasion front were not included in the criteria of Spiro. Therefore, however is not necessary.
· “In all malignant trumors… validated grading systems exist;” Do you mean “among all” SGCs?
· “The Past and Present of SGCs.” I cannot easily discern the past: is it that “WHO classification detached (sic) grades from tumor names”? Please specify.
· In the paragraph “The histological types” there is a messing of epidemiology (“the most prevalent malignancies”), with the role of the histological diagnosis per se, and grading of certain SGCs on prognosis. The authors give the impression that they do not agree with the fact that “WHO classification detached (sic) grades from tumor names”. Is it correct?
· The paragraph on Dedifferentiation should be restructured, as the definition-description of this phenomenon is followed by a comment on its presence in soft tissue tumors (?); later, the authors state “This transformation is reported not only in variable low-grade malignant tumors (acinic cell carcinoma…” but the characterization of various tumor subtypes’ grades does not fully agree with the information in Table 2, i.e., MEC is “variable grade” in Table 2 and “low-grade” in the text. Please rephrase.
· In the paragraph of micropapillary pattern there is a messing of a breast tumor, with a phenomenon observed in various adenocarcinomas, while the sentence “Among SGCs, the most prevalent are SDC; nonetheless, AdCC and intraductal papillary mucinous neoplasm (IPMN) were also reported.” Is incomplete. Was the presence of this feature associated somehow with prognosis in those SGCs? Please rephrase.
· Many of the features included in the “other histological features” have been previously presented in the section on dedifferentiation.
· “The other factors” includes parameters, like age and location, that (as the authors notice” are not histopathological. They could be mentioned in the Introduction as they are not strictly related to the Aim.
· The section “Validated grading systems for individual SGCs and similar diseases” begins with a summary of the content. Why didn’t a summary appear (for reasons of consistency) in previous paragraphs? In addition, what are the “similar diseases”? Aren’t they SGCs?
· Considering MEC grading the authors state that “MEC should be graded into low-grade, intermediate-grade, and high-grade groups”, but a few lines later “Nevertheless, high-grade MEC is very rare, and there is no difference in outcome between low and intermediate grades using any grading system”. The last sentence lacks appropriate references. What do they suggest after their review? Is it useful to use the AFIP or the Brandwein system, or no system at all?
· Considering NOS, the authors conclude that “Although this classification (profoundly by Spiro et al) reflected the tumor prognosis at that time, it is still somewhat reasonably useful today; however, it should be revised due to the difference in diagnostic criteria at present.” It is not clear whether WHO adopts this classification and/or this sentence reflects the opinion of the authors of the manuscript.
· The paragraph “Future Perspectives on SGCs” is not well-written, i.e., the terms “detectible” and “detectable” are used improperly, while, to my opinion, the information given on the techniques mentioned is well-known, and there is no need to be repeated.
· In the paragraph “Androgen receptor (AR)/ NK3 homeobox 1 (NKX3.1)” the concluding sentence is “On the other hand, the prognosis of AR, AMACR, or PAP-negative SDC was poor”. This has nothing to do either with the title/aim of the manuscript, as has already been said, or to the content of the paragraph that refers to drug treatment.
· “Presently, numerous protein inhibitors are being developed for various tumors, and SGCs are expected to be effective because some SGCs have genetic alterations in tumorigenesis”. Please rephrase.
· Why are TILs and TMB included in the same paragraph, as there seem to be separate processes? What is the meaning of the expression “global SGCs”?
Author Response
Dear Reviewer 2:
Thank you for your review and positive feedback. Please find below our point-by-point response to each comment. All revisions in the manuscript have been highlighted in yellow.
Comment: Aim states “We summarize the present SGCs classification…” This should not be included in the Aim as the presentation of the current classification of SGCs (in fact only of them, not all salivary gland tumors as in the manuscript) it is a prerequisite for the review.
Response: Thank you for this comment. We have revised the text accordingly.
Comment: The information included in the section “Future Perspectives on SGCs” is interesting but should be restructured in case that it should be included in a revised version of the manuscript. I.e., how those genetic alterations have been utilized in the diagnosis/treatment of other malignancies, how common are in SGCs and is specific subtypes, information on what has been done in the treatment of SGCs. This structure should be followed in all paragraphs.
Response: Thank you for your comment. Based on your comment, we have revised this section in the manuscript.
Comment: Due to numerous grammatical or syntaxis errors, the manuscript lacks consistency and is cannot be easily understood.
Response: Thank you for raising this point. The manuscript has been edited twice by a professional English editor at Editage to improve the overall readability and address any grammatical or syntax errors.
Comment: “The salivary gland comprises three major salivary glands (parotid, submandibular, and sublingual glands) and minor salivary glands”. Please rephrase.
Response: Thank you for your recommendation. We have accordingly revised the text.
Comment: “Salivary gland cancers (SGCs) arise from any salivary glands…”. SGCs (why not adenocarcinomas?) arise from any salivary gland (not plural form). Please rephrase.
Response: Thank you for your suggestion. In Table 1, salivary gland cancers refer to malignant tumors arising in the salivary gland. We have rephrased the text to reflect this.
Comment: “…the morbidity is approximately 10 to 20%”. What those percentages stand for, incidence, prevalence? Please specify.
Response: Thank you for raising this point. We have accordingly added the morbidity “rate.”
Comment: “SGCs show various tumor types”. What do you mean? Please rephrase.
Response: We apologize for this unclear phrasing and have revised the text for clarity.
Comment: “…due to the normal salivary gland consisting of inner luminal/epithelial or acinar/mucous cells and outer basal/myoepithelial cells in the duct or the secretory part.” Although I can think of what the authors intent to state, the sentence should be restructured.
Response: Thank you for your comment. We have revised the text to improve clarity and readability.
Comment: “The histopathological features correlate with the tumor progression, including lymph node/distant metastasis or local recurrence.” What are those “histopathological features”? Do you mean the histopathological subtypes?
Response: Thank you for your comment. Your interpretation was accurate. The text has been revised to better reflect this.
Comment: “A relatively long time ago, Spiro reported that the significant prognostic factors were the site of origin, histologic subtype, histologic grading, and clinical stage [4]. However, some carcinomas show mild cytological atypia, or evaluating the tumor invasion is challenging.” It is profound that cytological atypia and tumor invasion front were not included in the criteria of Spiro. Therefore, however is not necessary.
Response: Thank you for pointing this out. However, “histologic grading” in the report by Spiro included cytological atypia so we have opted to retain this sentence.
Comment: “In all malignant trumors… validated grading systems exist;” Do you mean “among all” SGCs?
Response: Thank you for your suggestion. The text has been revised accordingly.
Comment: “The Past and Present of SGCs.” I cannot easily discern the past: is it that “WHO classification detached (sic) grades from tumor names”? Please specify.
Response: Thank you for raising this point. We have revised the section heading for clarity.
Comment: In the paragraph “The histological types” there is a messing of epidemiology (“the most prevalent malignancies”), with the role of the histological diagnosis per se, and grading of certain SGCs on prognosis. The authors give the impression that they do not agree with the fact that “WHO classification detached (sic) grades from tumor names”. Is it correct?
Response: Thank you for raising this point. We agree with the WHO classification.
Comment: The paragraph on Dedifferentiation should be restructured, as the definition-description of this phenomenon is followed by a comment on its presence in soft tissue tumors (?); later, the authors state “This transformation is reported not only in variable low-grade malignant tumors (acinic cell carcinoma…” but the characterization of various tumor subtypes’ grades does not fully agree with the information in Table 2, i.e., MEC is “variable grade” in Table 2 and “low-grade” in the text. Please rephrase.
Response: Thank you for raising this point. We have included a more detailed description of MEC.
Comment: In the paragraph of micropapillary pattern there is a messing of a breast tumor, with a phenomenon observed in various adenocarcinomas, while the sentence “Among SGCs, the most prevalent are SDC; nonetheless, AdCC and intraductal papillary mucinous neoplasm (IPMN) were also reported.” Is incomplete. Was the presence of this feature associated somehow with prognosis in those SGCs? Please rephrase.
Response: Thank you for your suggestion. We have included a description on the prognosis of AdCC and IPMN.
Comment: Many of the features included in the “other histological features” have been previously presented in the section on dedifferentiation.
Response: Thank you for your suggestion. However, these features are independent prognostic factors of NOT dedifferentiated carcinomas.
Comment: “The other factors” includes parameters, like age and location, that (as the authors notice” are not histopathological. They could be mentioned in the Introduction as they are not strictly related to the Aim.
Response: Your suggestion is correct. But, as pathologists, we recognize that these parameters are also very important for histopathological diagnosis and have, thus, opted to include them in the main text.
Comment: The section “Validated grading systems for individual SGCs and similar diseases” begins with a summary of the content. Why didn’t a summary appear (for reasons of consistency) in previous paragraphs? In addition, what are the “similar diseases”? Aren’t they SGCs?
Response: Thank you for your comment. We described overall histological prognostic factors in the previous paragraph and each SGC in “Validated grading systems for individual SGCs and similar diseases”. The “similar diseases” included benign tumors, for example, pleomorphic adenoma (PA). Nearly all PA are benign, however, some exhibit clinical malignant progress (metastasizing PA) despite histologically benign morphology.
Comment: Considering MEC grading the authors state that “MEC should be graded into low-grade, intermediate-grade, and high-grade groups”, but a few lines later “Nevertheless, high-grade MEC is very rare, and there is no difference in outcome between low and intermediate grades using any grading system”. The last sentence lacks appropriate references. What do they suggest after their review? Is it useful to use the AFIP or the Brandwein system, or no system at all?
Response: Thank you for your comment. So have added appropriate references. These are all used clinically. These grading systems are all used clinically, and we added it.
Comment: Considering NOS, the authors conclude that “Although this classification (profoundly by Spiro et al) reflected the tumor prognosis at that time, it is still somewhat reasonably useful today; however, it should be revised due to the difference in diagnostic criteria at present.” It is not clear whether WHO adopts this classification and/or this sentence reflects the opinion of the authors of the manuscript.
Response: Thank you for your comment. The WHO classification does not clearly adopt it. We have revised the statement to clarify this point.
Comment: The paragraph “Future Perspectives on SGCs” is not well-written, i.e., the terms “detectible” and “detectable” are used improperly, while, to my opinion, the information given on the techniques mentioned is well-known, and there is no need to be repeated.
Response: Thank you for your comments. We have revised the text accordingly, with a particular emphasis on the proper usage of “detectible” and “detectable.” We have described the techniques for the reader less familiar with genetics.
Comment: In the paragraph “Androgen receptor (AR)/ NK3 homeobox 1 (NKX3.1)” the concluding sentence is “On the other hand, the prognosis of AR, AMACR, or PAP-negative SDC was poor”. This has nothing to do either with the title/aim of the manuscript, as has already been said, or to the content of the paragraph that refers to drug treatment.
Response: Thank you for your suggestion. We have added a brief description of androgen deprivation therapy for SGCs.
Comment: “Presently, numerous protein inhibitors are being developed for various tumors, and SGCs are expected to be effective because some SGCs have genetic alterations in tumorigenesis”. Please rephrase.
Response: We have revised the text for clarity.
Comment: Why are TILs and TMB included in the same paragraph, as there seem to be separate processes? What is the meaning of the expression “global SGCs”?
Response: Thank you for your comment. The TMB itself is not related to the prognosis, but TILs are influenced by TMB. Additionally, “global SGCs” refers to all SGCs; we have rephrased for clarity.

Reviewer 3 Report
Comments to the Authors
In this review article the authors summarized the main characteristics of salivary gland tumors focusing on the histopathological details and their impact on the disease prognosis. It also describes potential molecular targets in salivary gland carcinomas. The article is well organized, and authors performed a good job synthesizing the literature.
A few clarifications would further improve the manuscript:
1.- Please correct MED in line 83
2.-Add the citation for table number 2, if the table was built by the authors, please specify the criteria for this categorization. Of note, acinic cell carcinoma and PAC have metastasized potential and are in the low-grade category
3.- Under the headline “other histologic findings” the authors include local recurrence and distant metastasis but not specifically related to a histopathologic prognostic factor.
4.- In line number 180 please clarify the percentage of malignancy by location example: The risk of malignancy is variable: tumors arising in the parotid gland had a 25% risk of malignancy, instead sublingual tumors had a 70 to 90%. For tumors located in the minor salivary gland the risk is 50-75% (Jatin shah’s head and neck surgery and oncology fifth edition)
5.- Please add citations in line number 213 “there is no difference in outcome between low and intermediate grades using any grading system”
6.- Under the heading androgen receptor, information about the prevalence of AR positivity should be added. Also, I recommend discussing some therapeutic options for these patients (Dalin, M. G., et al. (2017). Androgen Receptor Signaling in Salivary Gland Cancer." Cancers (Basel) Boon, E., et al. (2018). "A clinicopathological study and prognostic factor analysis of 177 salivary duct carcinoma patients from The Netherlands." Int J Cancer)
7.- In the paragraph dedicated to discussing Tumor immunity/Tumor-infiltrating lymphocytes (TILs) potential targets relevant in other types of cancer are enumerated without a clear implication on salivary gland carcinoma, thus this section should be reformulated or erased from the manuscript.
Author Response
Dear Reviewer 3:
Thank you for your review and positive feedback. Please find below our point-by-point response to each comment. All revisions in the manuscript have been highlighted in yellow.
Comment: Please correct MED in line 83.
Response: Thank you for your comment. We have revised the text accordingly.
Comment: Add the citation for table number 2, if the table was built by the authors, please specify the criteria for this categorization. Of note, acinic cell carcinoma and PAC have metastasized potential and are in the low-grade category.
Response: Thank you for your comment. Citations have been added in the revised manuscript.
Comment: Under the headline “other histologic findings” the authors include local recurrence and distant metastasis but not specifically related to a histopathologic prognostic factor.
Response: Thank you for your comment. We have revised the text to address this aspect.
Comment: In line number 180 please clarify the percentage of malignancy by location example: The risk of malignancy is variable: tumors arising in the parotid gland had a 25% risk of malignancy, instead sublingual tumors had a 70 to 90%. For tumors located in the minor salivary gland the risk is 50-75% (Jatin shah’s head and neck surgery and oncology fifth edition)
Response: Thank you for raising this point. We have revised the text to include the percentage of malignancy by location and have added an appropriate citation.
Comment: Please add citations in line number 213 “there is no difference in outcome between low and intermediate grades using any grading system”
Response: Based on your comments, appropriate references have been added.
Comment: Under the heading androgen receptor, information about the prevalence of AR positivity should be added. Also, I recommend discussing some therapeutic options for these patients (Dalin, M. G., et al. (2017). Androgen Receptor Signaling in Salivary Gland Cancer." Cancers (Basel) Boon, E., et al. (2018). "A clinicopathological study and prognostic factor analysis of 177 salivary duct carcinoma patients from The Netherlands." Int J Cancer)
Response: Thank you for your suggestion. We have added a brief description of androgen deprivation therapy for SGCs.
Comment: In the paragraph dedicated to discussing Tumor immunity/Tumor-infiltrating lymphocytes (TILs) potential targets relevant in other types of cancer are enumerated without a clear implication on salivary gland carcinoma, thus this section should be reformulated or erased from the manuscript.
Response: Thank you for your suggestion. TILs may have a relationship with new drugs, such as metformin, in the future; we have accordingly modified the text to clarify this and have included relevant citations.

Round 2
Reviewer 2 Report
The authors have made the rquired corrections